# Multi-Time-Point Fecal Sampling in Human and Mouse Reveals the Formation of New Homeostasis in Gut Microbiota after Bowel Cleansing

**DOI:** 10.3390/microorganisms10122317

**Published:** 2022-11-23

**Authors:** Mingyang Li, Weike Qian, Leilei Yu, Fengwei Tian, Hao Zhang, Wei Chen, Yuzheng Xue, Qixiao Zhai

**Affiliations:** 1State Key Laboratory of Food Science and Technology, Jiangnan University, Wuxi 214122, China; 2School of Food Science and Technology, Jiangnan University, Wuxi 214122, China; 3National Engineering Research Center for Functional Food, Jiangnan University, Wuxi 214122, China; 4Wuxi Translational Medicine Research Center, Jiangsu Translational Medicine Research Institute Wuxi Branch, Wuxi 214122, China; 5Department of Gastroenterology, Affiliated Hospital of Jiangnan University, Wuxi 214122, China

**Keywords:** colonoscopy, bowel preparation, 16S rRNA gene, microbiome

## Abstract

Polyethylene glycol (PEG) is one of the most commonly used bowel cleansing methods. Although the safety of PEG for bowel cleansing has been proven, its impact on intestinal microbiota has not been clearly explained, especially in terms of the dynamic changes in intestinal microbiota after PEG bowel cleansing, and there are no consistent results. In this study, stool samples were collected from 12 participants at six time points before and after bowel cleansing. We obtained data on the microbiota of these samples using 16S rRNA gene sequencing and analysis. The data revealed that the structure and composition of the microbiota changed greatly approximately 7 d after intestinal cleansing. The analysis of the dynamic changes in the microbiota showed that the change was most significant at day 3, but the internal structure of the microbiota was similar to that before bowel cleansing. A comparison of the most significantly changed microbiota at different time points before and after bowel cleansing revealed four bacteria: Bacteroides, Roseburia, Eubacterium, and Bifidobacterium. We also established a humanized mouse model to simulate human bowel cleansing using PEG. The results showed that the mouse model achieved similar effects to human bowel cleansing, but its recovery speed was one stage earlier than that of humans. These findings suggest that the intestinal microbiota after bowel cleansing initially underwent a short-term change and then actively returned to its initial status. The results on key bacteria and establishment of mouse models can provide a reference for subsequent research on bowel cleansing.

## 1. Introduction

Both general anesthetic surgery and endoscopic tests require bowel cleansing. Among the various bowel-cleansing products, polyethylene glycol (PEG) supplemented with electrolytes is the most widely used in clinical practice [1,2]. Although the safety of PEG has been studied extensively, its effect on intestinal flora has not been studied sufficiently. When a large amount of PEG solution is passed through the intestines, many microorganisms are removed from the system. Therefore, the composition of gut microbiota changes significantly during bowel cleansing. Bowel cleansing with PEG can cause a series of intestinal symptoms, including diarrhea, intestinal gas, and abdominal pain [3]. According to a survey by Bini et al., 5.4% of subjects had persistent abdominal pain for 30 d after bowel cleansing [4]. In addition, a survey by Ko et al. pointed out that 11% of the subjects had abdominal discomfort 7 and 30 d after colonoscopy, and 94% of the subjects took 1–2 d off work because of mild discomfort symptoms [5]. These findings suggest that discomfort after bowel cleansing reduces patient satisfaction and increases social cost. We believe that the discomfort may be closely related to the disappearance of a large number of intestinal microbiota in the process of bowel cleansing.

To preserve mutualistic connections, the host and intestinal microbiota must be in equilibrium. The intestinal microbiota cannot maintain a balanced system with strong stimulation, such as antibiotic treatments, acute intestinal infections, dietary stimulation, or other external elements. Gradually, the gut microbiota establishes a new homeostasis. However, some early bacteria are lost, and some new ones are introduced during the establishment of the new homeostasis [6]. Among the common themes of recent studies on the human gut microbiome is the effect of diet on gut microbiota composition and, ultimately, host physiology [7]. A similar effect has been observed upon bowel cleansing with PEG. Some studies have found significant differences in the diversity or composition of fecal microorganisms before and after high-volume lavage [8,9,10,11] This suggests that bowel cleansing could adversely affect the composition and structure of the intestinal microbiota.

Several studies have shown modest associations between bowel cleansing and the microbiome. When analyzing the fecal and colonic mucosal microbiota after bowel cleaning, some studies found a considerable decrease in bacterial load and alpha diversity [8,12,13]; however, this has not always been proven [9,14]. Therefore, there is no consensus on how bowel cleansing affects the gut microbiota. Regarding the composition of the intestinal microbiota, the relative abundances of Bacteroidetes and Proteobacteria increase, and the relative abundance of Firmicutes decreases [3,12,15]. However, other studies have found no significant effects on microbiota after bowel cleansing [8,9]. No consensus has been established on how long bowel cleansing affects bacteria in the gut. One of these studies showed a long-lasting (28 d) effect on the composition and equilibrium of the gut microbiota [12]. Some studies have reported that the gut microbiota requires a month to recover [10,13,15]. Nevertheless, other studies have shown faster recovery of the microbiome [8,9]. The lack of analytical depth, inclusion of all kinds of people (healthy and unhealthy individuals), and limited level of sequencing have been the main reasons for the inconsistency in conclusions. In our research, we set the health criteria of the participants and used in-depth analysis methods. We tracked six time points to observe the dynamic changes in the microbiota. We wanted to find the key bacteria in the restoration of the gut environment and hoped to provide more feasible options to ease the discomfort of post-bowel cleansing. The results indicated that partial perturbations of the flora persisted for 28 d after bowel cleansing and were most pronounced in the early stages. In addition, our study established and validated a feasible humanized mouse model for the study of bowel cleansing that aims to simulate human bowel cleansing behavior.

## 2. Materials and Methods

### 2.1. Human Subjects

Twelve subjects [four men and eight women; mean age 43.8 years; mean body mass index (BMI), 21.6; Table 1] who were scheduled to undergo bowel preparation were recruited from China Wuxi. The baseline characteristics are shown in the Table 1. This study was approved by the Medical Ethics Committee of Jiangnan University (JNU20220606IRB06) and was implemented in accordance with the provisions of the Declaration of Helsinki. None of the test subjects was vegetarian, one was taking metformin hydrochloride, one was taking folic acid tablets, and two had taken antibiotics within the last three months. No medication was stopped during the study, in line with the ethical committee’s requirements. The remaining subjects had no underlying diseases or had been treated with antibiotics, immunosuppressants, or antacids during the preceding three months. Written informed consent was obtained from all participants involved in this study. This study was registered in the Chinese Clinical Trial Registry (ChiCTR2200064097).

### 2.2. Study Design and Sample Collection

The purpose of the study was to investigate the effects of taking the polyethylene glycol electrolyte lavage solution (PEG4000 60 g/L, sodium chloride 1.46 g/L, sodium sulfate 5.68 g/L, potassium chloride 0.74 g/L, and sodium bicarbonate 1.68 g/L, Jiangxi Hengkang Medicine Co. Ltd., Shangrao, Jiangxi, China) on the intestinal microbiota. Twelve people received a split dose of 3 L PEG before colonoscopy for bowel cleansing. Colonoscopy was performed at the Affiliated Hospital of the Jiangnan University, China. Samples were collected from the participants at six time points. These included a baseline sample donated the day before bowel cleansing (day 0), a sample immediately at the first defecation after bowel cleansing (day 3), and four follow-up samples 7, 14, 21, and 28 d after bowel cleansing (Figure 1). At each sample before and after the treatment, we recorded their diet. We excluded those with large changes in diet before and after bowel cleansing. Samples collected at home were immediately frozen in domestic freezers at −20 °C and were delivered to the endoscopy center as soon as possible where they were stored at −80 °C.

### 2.3. 16S rRNA Gene Sequencing and Analysis

DNA was extracted from the samples according to the Earth Microbiome Project (EMP) standard protocols using the FastDNA Spin Kit (MP Biomedicals Ltd., Santa Ana, CA, USA). Two hypervariable regions (V3 and V4) of the 16S rRNA gene sequence were amplified from the extracted DNA using 341F/806R primers and the DNA Gel/PCR Purification Miniprep Kit (Biomiga Ltd., Hangzhou, Zhejiang, China). Libraries were sequenced using the Illumina MiSeq platform (Illumina, Inc., San Diego, CA, USA). The DADA2 package from QIIME2 was used to perform quality filtering and demultiplexing, and the reads were assigned to open reference Amplicon sequence variants (ASVs). Sequence alignment was performed using the Silva Bacterial Database. The resulting ASV counts per sample were rarefied to 5625. We analyzed diversity with QIIME2 using the Qiime diversity core plugin. We used the R vegan, ape, ggpubr, and ggplot2 libraries for analysis and visualization, and Python’s NumPy and SciPy libraries for ASV-tracking. Sankey plots were drawn using the website (http://sankey-diagram-generator.acquireprocure.com/ (accessed on 28 July 2022)). The parameters of the random forest were determined using R random forest libraries.

### 2.4. Establishing the Humanized Mouse Model

Four-week-old (12.0 ± 2.0 g) specific pathogen-free (SPF) C57BL/6J male mice (Zhejiang Vital River Laboratory Animal Technology Co. Ltd., Hangzhou, Zhejiang, China) were used in the experiments. The mice were used to optimize gut microbiota transplant [16]. The specific experimental process is shown in Figure 2. The mice were housed in a controlled room (approximately four per cage) with a 12 h daylight cycle and free access to food and water. Fresh fecal samples from healthy volunteers were mixed with an equal volume of phosphate-buffered saline containing 20% glycerol and stored at −80 °C until use. After a one-week adaptation, the mice were gavaged with antibiotics (ATB) containing vancomycin (100 mg/kg), ampicillin (200 mg/kg), metronidazole (200 mg/kg), and neomycin sulfate (200 mg/kg) once a day for seven days to clean the gut microbiota [17]. Fecal samples were thawed and suspended with an equal volume of PBS, then vortexed and centrifuged for the supernatant to fecal microbiota transplantation (FMT). Two hundred μL of fecal suspension was administered to mice by oral gavage for five days. The mice were then randomly divided into two groups: a control group and an experimental group. The experimental group was gavaged with 1.5 mL PEG electrolyte solution (PEG4000, 92.5 g/L; sodium chloride, 1.9 g/L; sodium sulfate, 7.4 g/L; potassium chloride 0.98 g /L; sodium bicarbonate 2.2 g/L; Jiangxi Hengkang Medicine Co. Ltd., Shangrao, Jiangxi, China) for bowel cleansing (five times, 0.3 mL every 30 min). The sample collection is shown in Figure 2. All experiments were approved by the Committee of Ethics of Jiangnan University, China (JN. No20220615c0800824[210]).

### 2.5. Humanized Mouse Data Analyses

Mouse fecal DNA was extracted and sequenced in the same way as for the human samples. All sequences were filtered and demultiplexed by DADA2 using QIIME2. Each sample was rarefied to 16,343 ASVs. The R vegan, ape, ggpubr, and ggplot2 libraries were used for analysis and visualization. LEfSe analysis was performed using the online LEfSe program based on a normalized genus-level classification table (https://huttenhower.sph.harvard.edu/galaxy/ (accessed on 28 July 2022)).

### 2.6. Statistical Analysis

Statistical analyses were carried out with R or GraphPad Prism 9.4.0 using independent and paired *t*-tests, and nonparametric tests, as necessary. Differences were considered statistically significant at *p* < 0.05. The *p*-values were represented in plots as follows: ns, *p* >0.05 (not significant, may not be indicated), * *p* < 0.05, ** *p* < 0.01, *** *p* < 0.001, and **** *p* < 0.0001.

## 3. Results

### 3.1. Bowel Cleansing Altered the Microbiota Composition

Microbial composition showed a large change over time after bowel cleansing with PEG. We measured the microbiota membership of 12 participants at six time periods with 16S ribosomal RNA (rRNA) sequencing and found that the alpha diversity of the bacterial community changed significantly in the early stages (days 3 and 7) of bowel cleansing (Figure 3A and Appendix A). The alpha diversity returned to baseline (before bowel cleansing, 0 d) between days 0 and 14. We applied principal coordinate analysis (PCoA) to the Bray-Curtis distances of samples from different days. The microbiota of the individual participants, when plotted by day, showed an apparent regularity over time. Although the composition of the microbiota between days 0 and 3 was statistically significant (*p* = 0.0023, Wilcoxon), there was little or no difference among days 0, 14, and 28 (*p* = 0.7242 and *p* = 0.2161, Wilcoxon; Figure 3B). The specific changes in the taxonomic composition of the intestinal microbiota differed over time (Figure 3C). The highest relative abundance of Bacteroides before and after bowel cleansing may be correlated with bowel cleansing.

### 3.2. Dynamic Changes of the Microbiota after Bowel Cleansing

To investigate the dynamic changes in the microbiota after intestinal cleansing with PEG, we performed a longitudinal analysis of ASVs using the 16s rRNA sequence. We first clustered ASVs at the phylum level. Almost all phyla exhibited fluctuating changes after bowel cleansing. These changes were the strongest on days 3 and 7. Bacteroidetes and Firmicutes, with the highest relative abundances, showed the most obvious changes. Although most phyla returned to their original levels on day 28, changes in Bacteroidetes and Firmicutes appeared to continue (Figure 4A). At the genus level, the top 10 genera in terms of relative abundance were chosen to create an area map. On day 3, similar to the phylum level, different bacterial genera showed large fluctuations in abundance. However, interestingly, the results showed that the genus level did not reach equilibrium 14 d after bowel cleansing; for instance, Fusobacterium and Blautia continued to decline even after 28 d (Figure 4B).

We tracked individual ASVs within different phylum levels to clearly visualize the dynamic changes in the gut microbiota and revealed the dynamic changes in Bacteroidetes and Firmicutes (Figure 4C). Some of the Firmicutes ASVs displayed time volatility, with 12.2% disappearing between days 0 and 3; out of those that disappeared, 85.4% reappeared at later time points (Figure 4C, upper panel). A greater proportion of Bacteroidetes ASVs exhibited similar dynamic changes. A smaller number of Bacteroidetes ASVs disappeared between days 0 and 3 (9.6%); out of those that disappeared, 96.4% reappeared at later time points. Additionally, Bacteroidetes had the highest number of ASVs on day 3, up to 11.8% (Figure 4C, bottom panel). However, by day 28, the newly added ASVs at previous time points were almost not retained for either Bacteroidetes or Firmicutes; the results of the overall ASVs were most similar to those at day 0.

### 3.3. Changes in Specific Microbiota after Bowel Cleansing

The longitudinal samples collected during bowel cleansing were grouped into three clusters representing the middle, late, and early stages of bowel cleansing. The microbiota on day 0, which was before bowel cleansing, was the most similar to that in the middle stage (D7, D14, and D21; Figure 5A). Comparing day 3 with day 0, we found statistically significant changes in the bacterial genera at this stage (Wilcoxon paired signed-rank test, *p* < 0.05). The highly variable genera were Bifidobacterium, Bacteroides, Roseburia, and Eubacterium. By day 7, the microbiota began to recover in the middle stage of bowel cleansing. Statistical analysis of the abundance of these four genera at different time points revealed that all genera recovered on day 7, except for Bifidobacterium (Wilcoxon paired signed-rank test, *p* > 0.05). However, according to the changes in the personal flora connected by the gray line, Roseburia and Eubacterium did not recover in some people (Figure 5B). The recovery of Bifidobacterium may occur on day 14 of the middle stage. On day 28, there was a late bowel-cleansing stage, and there were no statistically significant differences among the four genera compared to day 0 (Wilcoxon paired signed-rank test, *p* > 0.05). However, according to the trend of bacterial flora described by the median line, both Bifidobacterium and Bacteroides remained unchanged after day 28 and did not reach a stable trend (Figure 5B). The change in the trend of key bacteria at the early stage of bowel cleansing predicted by the machine learning model established by random forest was consistent with the trend of our experiment (Figure 5C).

### 3.4. The Response of the Humanized Mice Model to Bowel Cleansing

We established a humanized mouse model to simulate changes in human gut microbiota after bowel cleansing with PEG. As the changes in the human microbiota after bowel cleansing mainly occurred within 0–14 d, experiments involving the humanized mice model were conducted within 0–14 d after the PEG treatment. After FMT, the fecal microbiota of recipient mice and their donors were compared. A distinct distribution between SPF and humanized mice was verified by the Bray-Curtis distance, with the gut microbiota of the latter being very similar to that of their donor (Appendix A).

We compared the microbiota of mice from the two groups at four time points and found that the alpha diversity was significantly reduced on the first day after the PEG treatment (*p* = 0.0021, Welch’s *t*-test). On the third day, there was no significant difference in the alpha diversity between the experimental and control groups. However, on the 14th day, the value of the experimental group was slightly lower than that of the control group (Figure 6A). We also applied PCoA to the Bray–Curtis distances of samples from different groups and treatment times. We found that there was always a gap in beta diversity between the experimental and control groups over time (Figure 6B). Then, we compared the Bray–Curtis distance between the experimental and the control groups at different time points and found a similar regularity to alpha diversity (Figure 6C). At the genus level, the bacterial genera in the experimental and control groups also changed over time (Figure 6D). The abundance of Bacteroides, which had the highest proportion in the early stage, also increased after the PEG treatment (compared to the control group). Colon slices were used to determine whether there was any effect on the colon after the PEG treatment. The slices in the experimental and control groups exhibited similar characteristics. The crypts were closely arranged, the goblet cells were abundant, and the entire intestinal wall was intact, showing no obvious abnormalities in the treatment group (Figure 6E).

### 3.5. The Key Bacteria in Humanized Mice after Bowel Cleansing

We used LEfSe analysis of fecal microbiota to identify the key bacteria in humanized mice after the PEG treatment. We found that the microbiota of mice changed the most on the first day after the PEG treatment; therefore, we compared the changes in microbiota on the first day. Overall, the significantly changed microbiota appeared to be more abundant in humanized mice than in humans. LEfSe cladograms revealed the enrichment of several bacterial taxa after the PEG treatment (Figure 7A). We found that the PEG treatment increased the relative abundance of Bacteroides, Blautia, Anaerostipes, and Ruminococcus, and decreased the relative abundance of Lactobacillus, Parabacteroides, and many others (Figure 7B). A comparison of the microbiota of all days after the PEG treatment showed that Bacteroidetes changed the most at the beginning (day 1) and Firmicutes changed the most at the end (day 14, Appendix A). This indicated that key bacteria were replaced in the early, middle, and late stages after the PEG treatment. We analyzed four of the bacteria that were most relevant to human bowel cleansing. We found that Bacteroides, Roseburia, and Eubacterium showed significant changes on day 1, similar to those observed in the early stage of human bowel cleansing (*p* = 0.0036, *p* = 0.0360, *p* = 0.0407, Welch *t*-test; Figure 7C). However, no significant changes in Bifidobacterium were found in humanized mice (Figure 7C).

## 4. Discussion

In this study, we investigated the specific effects of bowel preparation with PEG on gut microbiota and established an animal model to simulate this process. To date, there is no consensus on the effect of bowel preparation on intestinal microbiota. It is well known that the bowel preparation process removes a large number of bacteria, thereby reducing the bacterial load [10]. However, there are differing opinions on how bowel preparation affects the structure of the intestinal microbiota. Some studies have shown a decrease in alpha diversity [9,11,18], while others have shown no effect of bowel preparation [8,12,13]. The beta diversity analysis showed that there was no obvious regularity between different time points [8,9,11]. To explore the specific impact of bowel preparation on the microbiota, it is necessary to analyze the composition of the microbiota at different levels. Powles et al. showed no significant changes from the phylum to the genus level after bowel preparation [9]. Two other studies demonstrated the same results [8,19]. The results obtained in other studies were different, which may be due to different PEG doses, time point analysis, and analysis techniques. In our study, to better observe the dynamic changes in the intestinal microbiota after bowel preparation, we used six time points for sampling analysis. We also tracked ASVs at six time points before and after bowel preparation to reveal dynamic changes in the microbiota after bowel preparation during recovery. Finally, considering the limitations of population experiments, it may be difficult to perform bowel preparation experiments in special populations. Our study also established a humanized mouse model to study the effect of bowel preparation, which lays a foundation for subsequent research.

Our observation that the consumption of PEG washes away intestinal contents and affects the structure of the entire gut microbiota corroborates the results of previous studies showing a 34.7-fold reduction in bacterial load [12], an impact on bacterial diversity [8] and changes in the composition of the microbiota [9]. Some studies have suggested that bowel preparation has no effect on the structure of gut microbiota [11,20]. This is attributed to different sampling time points and numbers of participants, leading to different results. Another explanation is that individual differences in the microbiota are greater than the impact of bowel preparation [18,19]. In our study, individual differences were also obvious, but we were still able to observe the overall changes in the microbiome caused by bowel preparation. We performed PCOA analysis on the Bray-Curtis distance and observed the effects of individual differences and bowel preparation in the PCo1 and PCo2 dimensions (Figure 3B). These results suggest that the effect of bowel preparation on the composition of the microbiota is greater in the early stages after bowel preparation. Therefore, 3–7 d is an important bacterial flora marker point after bowel preparation that can be targeted for intervention.

As microbiota balance is increasingly used as an indicator of human health, studying the resilience of microbiota is imperative. Gut microbiota resilience has traditionally been studied from the perspective of membership dynamics before and after perturbations [20]. By comparing the differences in microbiota composition at different time points, we identified a common changing pattern at the phylum level (Figure 4A), which is consistent with previous studies [12,13]. These studies indicate that following bowel preparation, the relative abundance of Bacteroidetes increases, while the relative abundance of Firmicutes decreases. Studies have shown that the ratio of Bacteroidetes to Firmicutes (F/B) is an indicator of an abnormal state in the human body, such as obesity [21], diabetes [22], and non-alcoholic fatty liver disease [23]. Barnes et al. found that F/B can maintain a balanced state that is very beneficial to intestinal health [24]. Therefore, it can be speculated that the temporary metabolic disturbance caused by bowel preparation has an effect on flora that is similar to that caused by metabolic diseases. Our study tracked changes in ASVs at different time points as a complement to studies on the dynamics of microbiota (Figure 4C). This result indicates that the internal structure of the microbiota tends to return to its original state after a disturbance. Our research shows that changes in the external environment can affect the composition of the microbiota, and the change pattern of each phylum may be different; however, its internal structure will actively return to its previous state after perturbation.

Having established that bowel preparation leads to changes in the structure of the microbiota, we must know how specific microbiota changes so that these results can be utilized to customize therapy or clinical studies. It has previously been shown that the recolonization of bacteria in the face of disturbances is sequential [25]. Therefore, according to the changes in microbiota at different time points, we divided the microbiota after bowel preparation into three stages: 0–7 d as the early stage, 7–28 d as the middle stage, and 28 d later as the late stage (Figure 5A). The microbiota that were recovered in the early stage were the first colonizers with an immediate response; the microbiota that underwent obvious changes in the middle stage were secondary colonizers, and their interaction with the microbiota that colonized in the early stage may be the basis for some subsequent changes in the bacteria; some microbiota were still changing in the late stage, which may be because the colonization of these microbiota continued. Our study showed that changes in the microbiota were most obvious in the early stage, which is also consistent with the results of previous studies. Therefore, considering the individual differences in the population samples, we used the Wilcoxon test for each person to make a pairwise comparison of the day 0 and day 3 microbiota differences. We compiled information on the representative microbiota before and after bowel cleansing, and noted that they are the microbiota that are upregulated or downregulated after bowel cleansing (Appendix A). Then we found that four species of bacteria were significantly changed (Figure 5B). Jalanka et al. also observed changes in the microbiota before and after bowel preparation at the genus level, but the genera obtained were different from ours [12]. Owing to the different time points of our sampling and the eating habits of the recruited people, it was normal to obtain different results. However, these results indicated that there were significant changes in the microbiota during the early stages. Interestingly, we found that the relative abundance of Bifidobacterium decreased 3 d after bowel preparation. This study provides a basis for future customized therapies with probiotic supplementation. D’Souza et al. conducted a probiotic intervention in people after bowel preparation and found that supplementation with Bifidobacterium can relieve discomfort after bowel preparation [26]. However, they used multi-strain probiotics and did not account for changes in the gut microbiota, which should be studied in the future. In addition, similar changes in microbiota occur in infectious diarrhea [27] and have been observed with antibiotics [28]. This means that changes in the microbiota may be consistent when faced with similar external disturbances. This may be due to the relatively similar colonization of the microbiota under similar growth conditions. We found that the microbiota stabilized after 28 d, which is better than antibiotic treatment, and with a faster recovery time [29]. This indicates that bowel preparation is a relatively mild external perturbation, and normally, the resilience of the human gut microbiota is sufficient to ensure that it quickly returns to a stable state. However, compared to day 0, the microbiota reached a different stable state after 28 d. Fassarella et al. proposed that a new homeostasis is formed as a result of the flora in the face of disturbances [6]. Bowel preparation also follows this pattern of change. We did not analyze gut fungi because they account for far less than bacteria in the gut. It may also be a valuable direction to study the changes in gut fungal composition after bowel cleaning. Our study provides new evidence for this pattern, which can provide interesting directions for subsequent research.

Owing to the convenience of animal models, we established a mouse model to simulate human bowel-preparation behavior. Tropini et al. conducted a PEG treatment experiment on humanized mice to explore changes in microbiota [30]. They gavaged mice with PEG that was appreciably milder than PEG-based bowel preparation to observe changes in their microbiota. We obtained the equivalent of the PEG-based solution required for human bowel preparation according to calculations and previous experience [17,31]. We found that the overall change trend of the microbiota structure after bowel preparation was very similar between mice and humans, but the rate of change in mice seemed to be faster compared to humans (Figure 6A–D); this is also reflected in the study of Tropini et al. [30]. This may be because the metabolic response of mice is faster than that of humans, and the changes in the intestinal microbiota of mice are about 3 days faster than humans. The results of the colon sections showed that intestinal damage caused by bowel preparation was not obvious in the mice. However, Tropini et al. indicated that PEG treatment disrupted the colonic environment in mice [30]. This may be due to the fact that the mouse species they used was different from ours, and the gut of germ-free mice may be relatively more fragile. As there was no obvious damage to the intestinal environment of humans after bowel preparation, our model can simulate the human one well. Owing to the small individual differences in the mouse group, we used LEfSe to compare different groups of mice and obtained more differential bacteria than in the human experiment. Mice can almost reflect the same early bacterial flora changes as humans (Appendix A). The results of previous studies also partially corresponded to those in mice [11,12,19]. However, we found that changes in Bifidobacterium in mice were different from those in humans (Figure 7C). The reason for this inconsistency may be that fewer Bifidobacterium colonize the mouse gut. We found a significant decrease in *Lactobacillus* in mouse microbiota (Figure 7B), which is consistent with the study of Jalanka et al. [12]. In fact, some researchers have used *Lactobacillus* as a treatment after bowel cleansing [26,32]. We think *Lactobacillus* can also be recommended as a probiotic to relieve the discomfort of post-bowel cleansing, indicating that the mouse model can provide us with more intervention ideas. These results indicate that the mouse model can better simulate the behavior of human bowel preparation; however, the recovery speed after bowel preparation is one stage earlier (about 3 days) than in humans. This provides a suitable animal model when conducting population experiments is inappropriate or inconvenient.

## 5. Conclusions

In conclusion, our results indicate that bowel preparation had a certain impact on the intestinal flora, and this effect was most obvious in the early stages. Our results provide new evidence for the impact of external perturbations on the gut microbiota. However, owing to the limited number of human participants in our study, the possible conclusions may not be representative of the changes in the microbiota after bowel preparation. Future studies with larger cohorts and different groups may prove this point. The key bacteria identified in our study may be useful for future microbiota studies. In addition, our animal model can be used in future studies on bowel preparation to reveal more general conclusions.

## Figures and Tables

**Figure 1 microorganisms-10-02317-f001:**
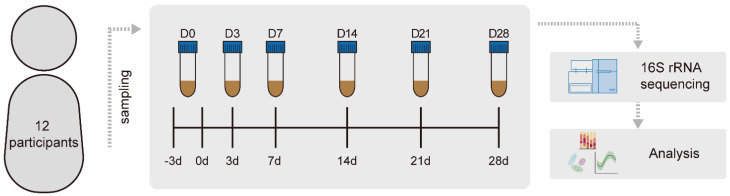
Schematic of population experimental design and collection protocol. Twelve participants were sampled at 6 time points before and after bowel cleansing.

**Figure 2 microorganisms-10-02317-f002:**
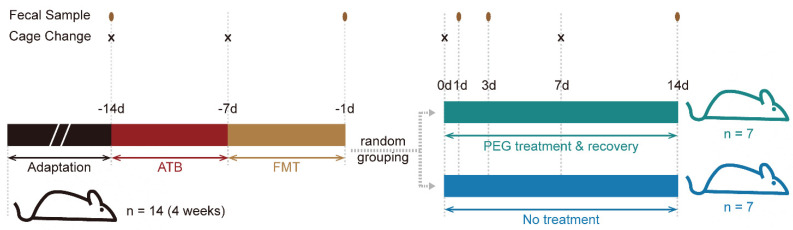
Schematic of animal experimental design and collection protocol. The adaptation period of mice is one week; ATB, gavage with antibiotics; FMT, fecal microbiota transplantation from healthy donor to mice; PEG treatment, gavage polyethylene glycol electrolyte solution for bowel cleansing. The times of fecal sample and cage change have been marked in the figure.

**Figure 3 microorganisms-10-02317-f003:**
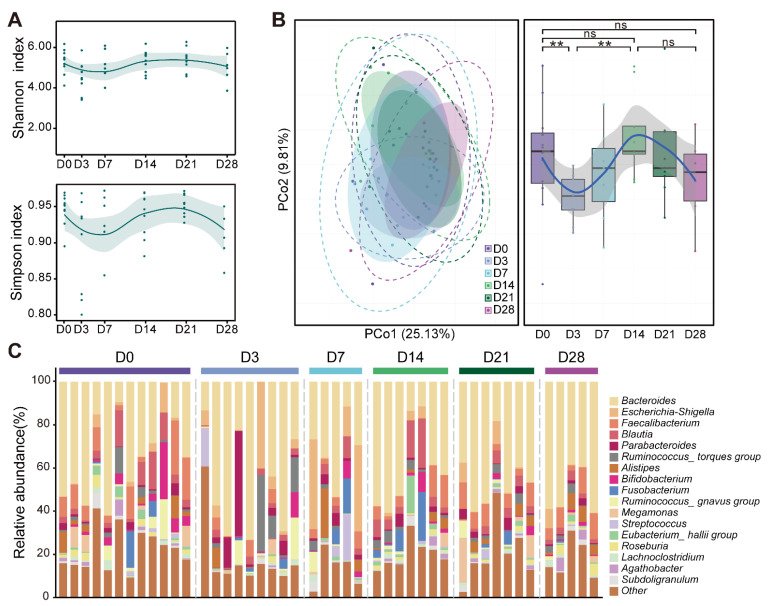
The structure of intestinal microbiota changed after bowel preparation. (**A**) The alpha diversity of the microbiota after bowel preparation over time. The top and bottom panels show the Shannon and Simpson indices of the microbiota, respectively. (**B**) Individual gut microbiota compositions draw a Bray-Curtis PCoA plot in different time groups (left panel). There are some differences in the composition of the flora at different times (right panel). Significance was measured using the Wilcoxon rank-sum test. Boxplot center values indicate the median value, error bars indicate the extreme value. The dotted ellipse borders represent the 95% confidence interval (CI), the shaded ellipses represent the 80% CI. (**C**) Percentage bar graph showing the relative proportion of bacterial genera in each sample. Curve fitting in all plots is conducted by the “LOESS” method. The shadow around the curve shows the 95% confidence interval. *p* > 0.05, ns (not significant, may not be indicated); ** *p* < 0.01.

**Figure 4 microorganisms-10-02317-f004:**
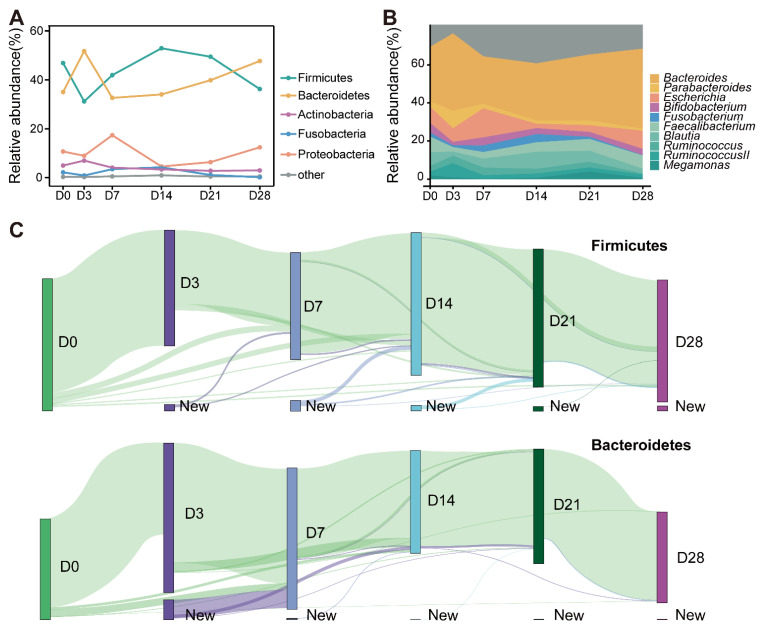
Dynamic changes in microbiota after bowel preparation at different levels. (**A**) Changes in different intestinal microbiota over time at the phylum level. (**B**) The area map shows the trend of the relative abundance of different bacterial groups at the genus level. The figure shows the 10 genera with the highest relative abundance, where the genera represented by similar colors correspond to the phyla in Figure 3A. (**C**) ASVs are tracked using Sankey plots in both Firmicutes and Bacteroidetes over time. The heights of the rectangle indicate the relative number of ASVs. The lines represent the transition of ASVs between different time points and are colored according to the time point of first appearance.

**Figure 5 microorganisms-10-02317-f005:**
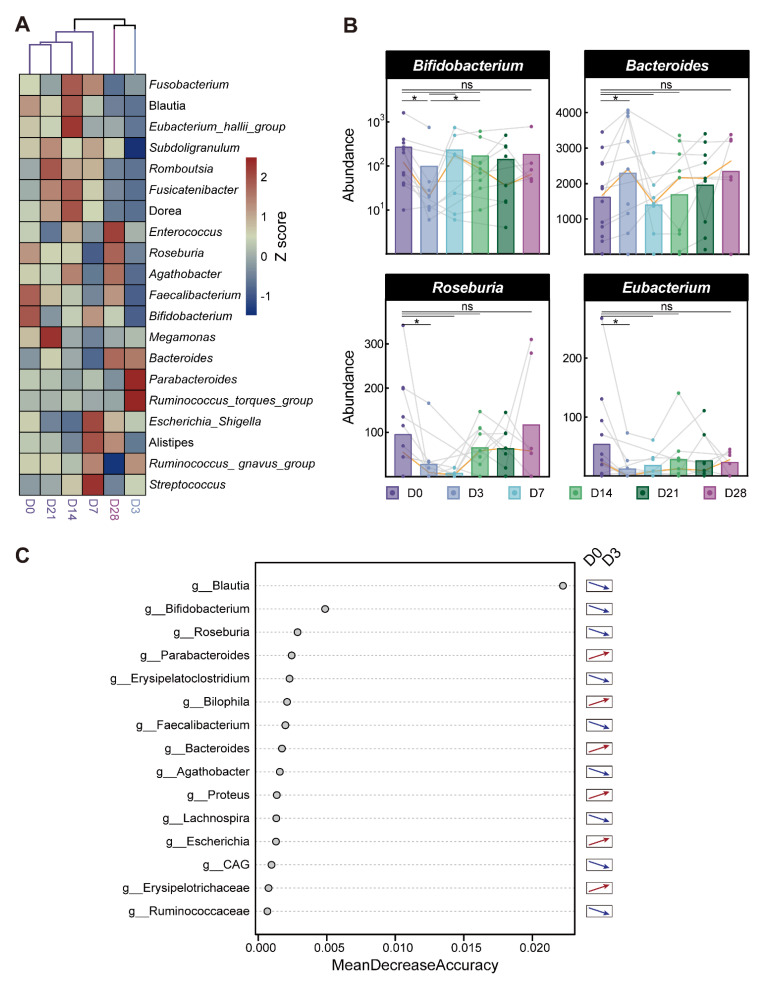
The specific microbiota changed with time after bowel preparation. (**A**) Heatmap showing clustering of intestinal microbiota at the genus level. (**B**) The change trend of four kinds of key bacteria over time was obtained using statistics. Each gray line represents one of the participants and the orange line represents the change in the median. Significance was measured using the Wilcoxon paired signed-rank test. (**C**) Random forest predicted the change trend of genus level microbiota in the early stage after bowel preparation. The out-of-bug (OOB) estimate of error rate is 12.5%. *p* > 0.05, ns (not significant, may not be indicated); * *p* < 0.05.

**Figure 6 microorganisms-10-02317-f006:**
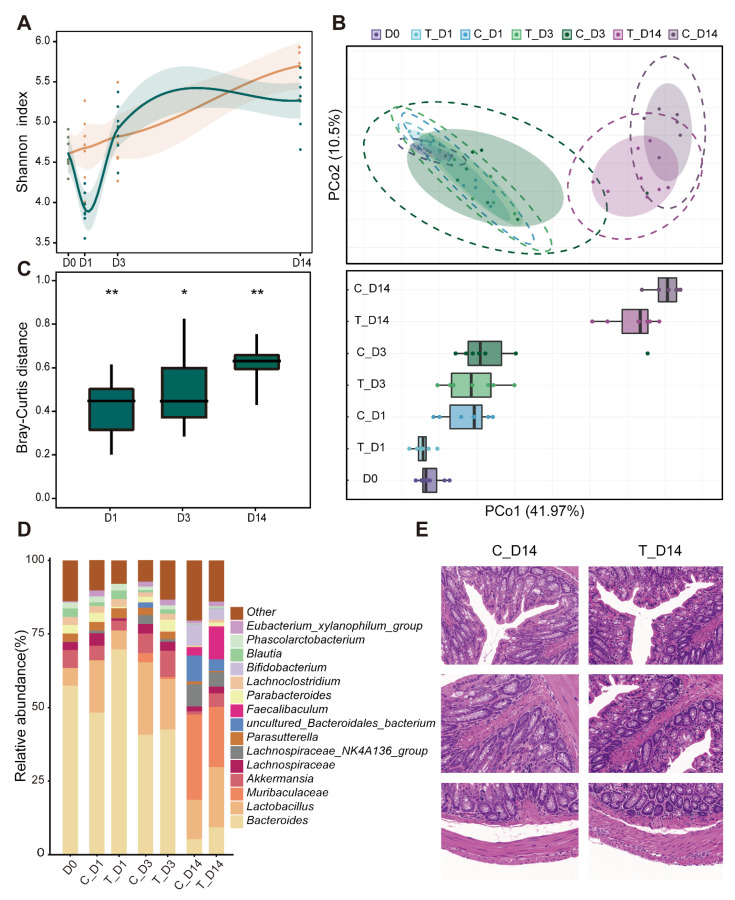
PEG treatment leads to corresponding changes in humanized mice. (**A**) Changes in the Shannon index of the experimental and control groups. Curve fitting is conducted by the “LOESS” method. The shadow around the curve shows a 95% confidence interval. (**B**) The Bray–Curtis PCoA plot shows the differences between different groups of mice at different time points (top panel). The microbiota composition of the different groups at each time point shows obvious differences (bottom panel). (**C**) The Bray–Curtis distance between microbiota of the experimental and control groups at every time point. (**D**) Percentage bar graph showing the relative proportion of bacterial genera in each group. (**E**) Comparison of HE-stained colon sections of the control group and the experimental group at day 14. *p* > 0.05, * *p* < 0.05; ** *p* < 0.01; T, polyethylene glycol (PEG) treatment group; C, control group.

**Figure 7 microorganisms-10-02317-f007:**
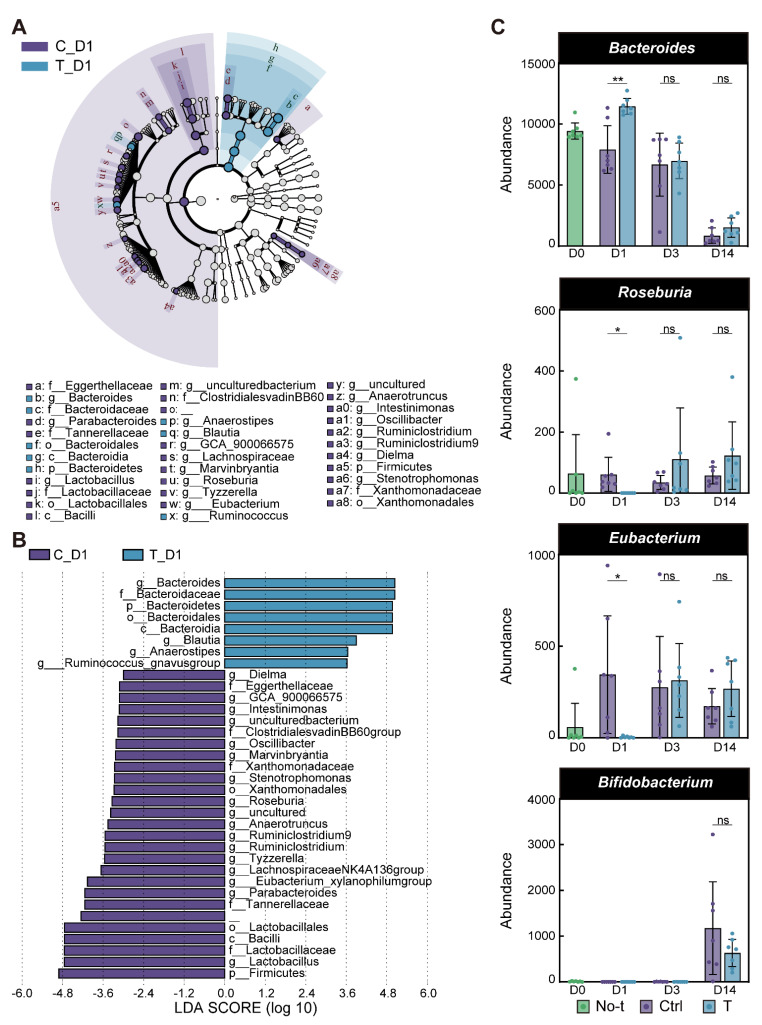
The key bacteria after the PEG treatment. (**A**,**B**) LefSe analysis was performed on the key bacteria that changed one day after colon cleansing; the LefSe cladograms A and column chart B show the results. (**C**) The change trend of Bacteroides, Roseburia, Eubacterium, and Bifidobacterium over time. Significance was measured using Welch *t*-test, and error bars show the 95% confidence interval. *p* > 0.05, ns (not significant, may not be indicated); * *p* <0.05; ** *p* <0.01; T, polyethylene glycol (PEG) treatment group; C, control group.

**Table 1 microorganisms-10-02317-t001:** Subject demographics.

Subject ID	Sex	Age	Height (cm)	Weight (kg)	BMI (kg/m^2^)	Number of Samples
1	F	52	170	68	23.5	4
2	F	34	162	50	19.1	5
3	M	28	173	70	23.4	5
4	F	59	163	61	23.0	4
5	M	49	173	71	23.7	5
6	M	53	173	57	19.0	5
7	F	60	165	60	22.0	2
8	F	22	158	47	18.8	4
9	F	22	163	58	21.8	4
10	M	58	170	60	20.8	4
18	F	43	160	56	21.9	5
19	F	45	172	65	22.0	5

M, male; F, female; BMI, Body Mass Index; Inclusion criteria for all subjects were: age between 18 and 65, BMI between 18.5 and 23.9. The number of samples is insufficient due to personal reasons. Subjects ID 11–15 withdrew from the group before experiment; 16 and 17 were removed for failing to take PEG electrolyte lavage solution as required.

## Data Availability

Not applicable.

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
