# Peer review of "Multi-Time-Point Fecal Sampling in Human and Mouse Reveals the Formation of New Homeostasis in Gut Microbiota after Bowel Cleansing"

_microorganisms, 2022, doi:10.3390/microorganisms10122317_

Round 1

Reviewer 1 Report

Dear authors

The article titled “Multi-time point fecal sampling in human and mouse reveals the formation of new homeostasis in gut microbiota after bowel cleansing” is interesting and describes the formation of new homeostasis in gut microbiota after a certain period of time after bowel cleansing. The authors have examined the short-term changes in the intestinal microbiota using 16S rRNA gene sequencing and analysis after bowl cleansing that actively returns to its initial status. Informative data about gut microbiome has been generated and analyzed properly by fecal analysis at multiple time points before and after bowel cleansing. Moreover, a humanized mouse model has been established for further studies on the human gut microbiome, but the manuscript should go under general refinement by adding some information mentioned below to make the concept more clear and understandable

1.      The introduction lacks a connection between paragraph number 1 and 2. Add a connective sentence to endure the connection between bowel cleansing and gut microbiota.

2.      In line 83, add the same information about female subjects as mentioned about man subjects and add the reference to Table 1.

3.      Is there any information available about the microbiota left in the gut after bowel cleansing? What percentage of gut microorganisms washed out during bowel cleansing? Add this information to give a clear picture of the effect of bowel cleansing on the commensal microbiome.  

4.      Mention the reason for not analyzing gut fungal composition in the study?

5.      Authors showed that the microbiota reached a different stable state after 28 days of the treatment (lines 490-491). Is this because of the difference in diet that subjects used before and after the treatment? Did subjects use a controlled diet throughout the experiment?

6.      The discussion part is written well. The authors suggested the use of probiotics to release the discomfort of post-bowel cleansing in the discussion part. Should you recommend the use of Lactobacillus-containing probiotics because the relative abundance of this microbe decreased according to your observation (line 345)?

Reviewer 2 Report

1. All PEG not used for bowel cleaning. Mention the specific category. Also mention the source of PEG.

2. THe rationale and objective is not clear.

3. What is the dose of PEG used for rats.

4. Fonts and Number in Figure not visible.

5. How the data of humans and rats are correlated from the study.

Reviewer 3 Report

The paper study the microbiota after bowel cleansing in six different time-points, analyzing human and mouse samples. It is fair designed and written. However, the low number of subjects analyzed may not be representative as mentioned in the conclusion. I have some concerns that I need to point.

1 – Line 39 “…has rarely been studied.” Well, actually there are a few papers published about this topic already. So, “rarely” is not appropriate.

2 – Line 85 says that the baseline characteristics are shown in the Supplementary table. However, there is no supplementary table provided by the authors.

3 – Line 96 says “This is a table…”. I believe someone forgot to add the title of the table and left the original template. In the table, the subject ID has 12 numbers from 1 to 10 then jumps to 18 and 19. Why? Describe better the missing ones

 4 – Line 130. The lack of information is concerning. The topic describes the humanized mouse model, and it should be very well described in this paper, not only giving refs.

5 - Figure 1B. I suggest separating the figure 1B from the figure 1 and create a figure 2, describing better the method in the legend, informing what is ATB, FMT, and PEG. The legend of all figures show explains all the information present in the graphics. Besides, the figure shows the adaptation period as Day -14, then ATB at Day -7, and at the end, FMT at Day -1. However, throughout the paper, it is used “#d” to describe the days. It must be standardized, choosing either “Day #” or “#d”.  

6 - It must be informed, even briefly, which antibiotic(s) was(were) used and the concentration(s).

7 - According to the reference provided by the authors (ref. 17), after the ATB treatment, mice were gavage with FMT. However, it is already evidenced that gavage mice with FMT right after ATB treatment is not appropriate. The reason is that ATB residues are still present in the GI tract in the transition phase ATB/FMT. This plays a role in the new microbiota provided in the FMT. For that reason, the correct way to work is giving 2 days gap (just providing autoclaved water) in between the ATB and the FMT administration. Could the authors explain why going from ATB to FMT without any intervals?

Round 2

Reviewer 2 Report

Accept